# Differentiable Ranks and Sorting using Optimal Transport

**Marco Cuturi    Olivier Teboul    Jean-Philippe Vert**
Google Research, Brain Team
{cuturi,oliviert,jpvert}@google.com

## Abstract

Sorting is used pervasively in machine learning, either to define elementary algorithms, such as $k$-nearest neighbors ($k$-NN) rules, or to define test-time metrics, such as top-$k$ classification accuracy or ranking losses. Sorting is however a poor match for the end-to-end, automatically differentiable pipelines of deep learning. Indeed, sorting procedures output two vectors, neither of which is differentiable: the vector of sorted values is piecewise linear, while the sorting permutation itself (or its inverse, the vector of ranks) has no differentiable properties to speak of, since it is integer-valued. We propose in this paper to replace the usual `sort` procedure with a differentiable proxy. Our proxy builds upon the fact that sorting can be seen as an optimal assignment problem, one in which the $n$ values to be sorted are matched to an *auxiliary* probability measure supported on any *increasing* family of $n$ target values. From this observation, we propose extended rank and sort operators by considering optimal transport (OT) problems (the natural relaxation for assignments) where the auxiliary measure can be any weighted measure supported on $m$ increasing values, where $m \neq n$. We recover differentiable operators by regularizing these OT problems with an entropic penalty, and solve them by applying Sinkhorn iterations. Using these smoothed rank and sort operators, we propose differentiable proxies for the classification 0/1 loss as well as for the quantile regression loss.

## 1   Introduction

Sorting $n$ real values stored in an array $\mathbf{x} = (x_1, \ldots, x_n) \in \mathbb{R}^n$ requires finding a permutation $\sigma$ in the symmetric group $\mathcal{S}_n$ such that $\mathbf{x}_\sigma := (x_{\sigma_1}, \ldots, x_{\sigma_n})$ is increasing. A call to a sorting procedure returns either the vector of sorted values $S(\mathbf{x}) := \mathbf{x}_\sigma$, or the vector $R(\mathbf{x})$ of the ranks of these values, namely the inverse of the sorting permutation, $R(\mathbf{x}) := \sigma^{-1}$. For instance, if the input vector $\mathbf{x} = (0.38, 4, -2, 6, -9)$, one has $\sigma = (5, 3, 1, 2, 4)$, and the sorted vector $S(\mathbf{x})$ is $\mathbf{x}_\sigma = (-9, -2, 0.38, 4, 6)$, while $R(\mathbf{x}) = \sigma^{-1} = (3, 4, 2, 5, 1)$ lists the rank of each entry in $\mathbf{x}$.

**On (not) learning with sorting and ranking.** Operators $R$ and $S$ play an important role across statistics and machine learning. For instance, $R$ is the main workhorse behind order statistics [12], but also appears prominently in $k$-NN rules, in which $R$ is applied on a vector of distances to select the closest neighbors to a query point. Ranking is also used to assess the performance of an algorithm: either at test time, such as $0/1$ and top-$k$ classification accuracies and NDCG metrics when learning-to-rank [21], or at train time, by selecting pairs [9, 8] and triplets [37] of points of interest. The sorting operator $S$ is of no less importance, and can be used to handle outliers in robust statistics, as in trimmed [20] and least-quantile regression [32] or median-of-means estimators [26, 25]. Yet, and although examples of using $R$ and $S$ abound in ML, neither $R$ nor $S$ are actively used in end-to-end learning approaches: while $S$ is not differentiable everywhere, $R$ is outright pathological, since it is piecewise constant and has therefore a Jacobian $\partial R / \partial \mathbf{x}$ that is almost everywhere zero.

**Everywhere differentiable proxies to ranking and sorting.** Replacing the usual ranking and sorting operators by differentiable approximations holds an interesting promise, as it would immediately enable an end-to-end training of any algorithm or metric that uses sorting. For instance, all of the test metrics enumerated above could be upgraded to training losses, if one were able to replace their inner calls to $R$ and $S$ by differentiable proxies. More generally, one can envision applications in which these proxies can be used to impose rank/sorting based constraints, such as fairness considerations that rely on the quantiles of (logistic) regression outputs [14, 22]. In the literature, such smoothed ranks operators appeared first in [36], where a softranks operator is defined as the expectation of the rank operator under a random perturbation, $\mathbb{E}_{\mathbf{z}}[R(\mathbf{x} + \mathbf{z})]$, where $\mathbf{z}$ is a standard Gaussian random vector. That expectation (and its gradient w.r.t. $\mathbf{x}$) were approximated in [36] using a $O(n^3)$ algorithm. Shortly after, [29] used the fact that the rank of each value $x_i$ in $\mathbf{x}$ can be written as $\sum_j \mathbf{1}_{x_i > x_j}$, and smoothed these indicator functions with logistic maps $g_\tau(u) := (1 + \exp(-u/\tau))^{-1}$. The soft-rank operator they propose is $A\mathbf{1}_n$ where $A = g_\tau(D)$, where $g_\tau$ is applied elementwise to the pairwise matrix of differences $D = [x_i - x_j]_{ij}$, for a total of $O(n^2)$ operations. A similar yet more refined approach was recently proposed by [18], building on the same pairwise difference matrix $D$ to output a unimodal row-stochastic matrix. This yields as in [36] a probabilistic rank for each input.

**Our contribution: smoothed $R$ and $S$ operators using optimal transport (OT).** We show first that the sorting permutation $\sigma$ for $\mathbf{x}$ can be recovered by solving an optimal assignment (OA) problem, from an input measure supported on all values in $\mathbf{x}$ to a second *auxiliary* target measure supported on *any* increasing family $\mathbf{y} = (y_1 < \cdots < y_n)$. Indeed, a key result from OT theory states that, pending a simple condition on the matching cost, the OA is achieved by matching the smallest element in $\mathbf{x}$ to $y_1$, the second smallest to $y_2$, and so forth, therefore "revealing" the sorting permutation of $\mathbf{x}$. We leverage the flexibility of OT to introduce generalized "split" ranking and sorting operators that use target measures with only $m \neq n$ weighted target values, and use the resulting optimal transport plans to compute convex combinations of ranks and sorted values. These operators are however far too costly to be of practical interest and, much like sorting algorithms, remain non-differentiable. To recover tractable and differentiable operators, we regularize the OT problem and solve it using the Sinkhorn algorithm [10], at a cost of $O(nm\ell)$ operations, where $\ell$ is the number of Sinkhorn iterations needed for the algorithm to converge. We show that the size $m$ of the target measure can be set as small as $3$ in some applications, while $\ell$ rarely exceeds $100$ with the settings we consider.

**Outline.** We recall first the link between the $R$ and $S$ operators and OT between 1D measures, to define then generalized Kantorovich rank and sort operators in §2. We turn them into differentiable operators using entropic regularization, and discuss in §3 the several parameters that can shape this smoothness. Using these smooth operators, we propose in §4 alternatives to cross-entropy and least-quantile losses to learn classifiers and regression functions.

**Notations.** We write $\mathbb{O}_n \subset \mathbb{R}^n$ for the set of increasing vectors of size $n$, and $\Sigma_n \subset \mathbb{R}_+^n$ for the probability simplex. $\mathbf{1}_n$ is the $n$-vector of ones. Given $\mathbf{c} = (c_1, \ldots, c_n) \in \mathbb{R}^n$, we write $\overline{\mathbf{c}}$ for the cumulative sum of $\mathbf{c}$, namely vector $(c_1 + \cdots + c_i)_i$. Given two permutations $\sigma \in \mathcal{S}_n, \tau \in \mathcal{S}_m$ and a matrix $A \in \mathbb{R}^{n \times m}$, we write $A_{\sigma\tau}$ for the $n \times m$ matrix $[A_{\sigma_i \tau_j}]_{ij}$ obtained by permuting the rows and columns of $A$ using $\sigma, \tau$. For any $x \in \mathbb{R}$, $\delta_x$ is the Dirac measure on $x$. For a probability measure $\xi \in \mathcal{P}(\mathbb{R})$, we write $F_\xi$ for its cumulative distribution function (CDF), and $Q_\xi$ for its quantile function (generalized if $\xi$ is discrete). Functions are applied element-wise on vectors or matrices; the $\circ$ operator stands for the element-wise product of vectors.

## 2 Ranking and Sorting as an Optimal Transport Problem

The fact that solving the OT problem between two discrete univariate measures boils down to sorting is well known [33, §2]. The usual narrative states that the Wasserstein distance between two univariate measures reduces to comparing their quantile functions, which can be obtained by inverting CDFs, which are themselves computed by considering the sorted values of the supports of these measures. This downstream connection from OT to quantiles, CDFs and finally sorting has been exploited in several works, notably because the $n \log n$ price for sorting is far cheaper than the order $n^3 \log n$ [35] one has to pay to solve generic OT problems. This is evidenced by the recent surge in interest for sliced Wasserstein distances [30, 5, 23]. We propose in this section to go instead *upstream*, that is to redefine ranking and sorting functions as byproducts of the resolution of an optimal assignment problem between measures supported on the reals. We then propose in Def.1 generalized rank and sort operators using the Kantorovich formulation of OT.

**Solving the OT problem between 1D measures using sorting.** Let $\xi, \upsilon$ be two discrete probability measures on $\mathbb{R}$, defined respectively by their supports $\mathbf{x}, \mathbf{y}$ and probability weight vectors $\mathbf{a}, \mathbf{b}$ as $\xi = \sum_{i=1}^{n} a_i \delta_{x_i}$ and $\upsilon = \sum_{j=1}^{m} b_j \delta_{y_j}$. We consider in what follows a *translation invariant* and *non-negative* ground metric defined as $(x, y) \in \mathbb{R}^2 \mapsto h(y - x)$, where $h : \mathbb{R} \to \mathbb{R}_+$. With that ground cost, the OT problem between $\xi$ and $\upsilon$ boils down to the following LP, writing $C_{\mathbf{xy}} := [h(y_j - x_i)]_{ij}$,

$$\mathrm{OT}_h(\xi, \upsilon) := \min_{P \in U(\mathbf{a}\,\mathbf{b})} \langle P, C_{\mathbf{xy}} \rangle, \text{ where } U(\mathbf{a}, \mathbf{b}) := \{ P \in \mathbb{R}_+^{n \times m} | P\mathbf{1}_m = \mathbf{a}, P^T\mathbf{1}_n = \mathbf{b} \}. \quad (1)$$

We make in what follows the additional assumption that $h$ is *convex*. A fundamental result [33, Theorem 2.9] states that in that case (see also [13] for the more involved case where $h$ is concave) $\mathrm{OT}_h(\xi, \upsilon)$ can be computed in closed form using the quantile functions $Q_\xi, Q_\upsilon$ of $\xi, \upsilon$:

$$\mathrm{OT}_h(\xi, \upsilon) = \int_{[0,1]} h\left(Q_\upsilon(u) - Q_\xi(u)\right) du. \quad (2)$$

Therefore, to compute OT between $\xi$ and $\upsilon$, one only needs to integrate the difference in their quantile functions, which can be done by inverting the empirical distribution functions for $\xi, \upsilon$, which itself only requires sorting the entries in $\mathbf{x}$ and $\mathbf{y}$ to obtain their sorting permutations $\sigma$ and $\tau$. Additionally, Eq. (2) allows us not only to recover the value of $\mathrm{OT}_h$ as defined in Eq. (1), but it can also be used to recover the corresponding optimal solution $P_\star$ in $n + m$ operations, using the permutations $\sigma$ and $\tau$ to build a so-called north-west corner solution [28, §3.4.2]:

**Proposition 1.** *Let $\sigma$ and $\tau$ be sorting permutations for $\mathbf{x}$ and $\mathbf{y}$. Define $N$ to be the north-west corner solution using permuted weights $\mathbf{a}_\sigma, \mathbf{b}_\tau$. Then $N_{\sigma^{-1}, \tau^{-1}}$ is optimal for (1).*

Such a permuted north-western corner solution is illustrated in Figure 1(b). It is indeed easy to check that in that case $(P_\star)_{\sigma, \tau}$ runs from the top-left (north-west) to the bottom right corner. In the simple case where $n = m$ and $\mathbf{a} = \mathbf{b} = \mathbf{1}_n / n$, the solution $N_{\sigma^{-1}, \tau^{-1}}$ is a permutation matrix divided by $n$, namely a matrix equal to $0$ everywhere except for its entries indexed by $(i, \tau \circ \sigma^{-1})_i$ which are all equal to $1/n$. That solution is a vertex of the Birkhoff [3] polytope, namely, an optimal assignment which to the $i$-th value in $\mathbf{x}$ associates the $(\tau \circ \sigma^{-1})_i$-th value in $\mathbf{y}$; informally, this solution assigns the $i$-th smallest entry in $\mathbf{x}$ to the $i$-th smallest entry in $\mathbf{y}$.

**Generalizing sorting, CDFs and quantiles using optimal transport.** From now on in this paper, we make the crucial assumption that $\mathbf{y}$ is already sorted, that is, $y_1 < \cdots < y_m$. $\tau$ is therefore the identity permutation. When in addition $n = m$, the $i$-th value in $\mathbf{x}$ is simply assigned to the $\sigma_i^{-1}$-th value in $\mathbf{y}$. Conversely, and as illustrated in Figure 1(a), the rank $i$ value in $\mathbf{x}$ is assigned to the $i$-th value $y_i$. Because of this, $R$ and $S$ can be rewritten using the optimal assignment matrix $P_\star$:

**Proposition 2.** *Let $n = m$ and $\mathbf{a} = \mathbf{b} = \mathbf{1}_n / n$. Then for all strictly convex functions $h$ and $\mathbf{y} \in \mathbb{O}_n$, if $P_\star$ is an optimal solution to (1), then*

$$R(\mathbf{x}) = n^2 P_\star \overline{\mathbf{b}} = n P_\star \begin{bmatrix} 1 \\ \vdots \\ n \end{bmatrix} = n F_\xi(\mathbf{x}), \quad S(\mathbf{x}) = n P_\star^T \mathbf{x} = Q_\xi(\overline{\mathbf{b}}) \in \mathbb{O}_n.$$

These identities stem from the fact that $n P_\star$ is a permutation matrix, which can be applied to the vector $n \overline{\mathbf{b}} = (1, \ldots, n)$ to recover the rank of each entry in $\mathbf{x}$, or transposed and applied to $\mathbf{x}$ to recover the sorted values of $\mathbf{x}$. The former expression can be equivalently interpreted as $n$ times the CDF of $\xi$ evaluated elementwise to $\mathbf{x}$, the latter as the quantiles of $\xi$ at levels $\overline{\mathbf{b}}$. The identities in Prop. 2 are valid when the input measures $\xi, \upsilon$ are uniform and of the same size. The first contribution of this paper is to consider more general scenarios, in which $m$, the size of $\mathbf{y}$, can be smaller than $n$, and where weights $\mathbf{a}, \mathbf{b}$ need not be uniform. This is a major departure from previous references [18, 36, 29], which all require pairwise comparisons between the entries in $\mathbf{x}$. We show in our applications that $m$ can be as small as 3 when trying to recover a quantile, as in Figs. 1, 3.

**Kantorovich ranks and sorts.** The so-called Kantorovich formulation of OT [33, §1.5] can be used to compare discrete measures of varying sizes and weights. Solving that problem usually requires *splitting* the mass $a_i$ of a point $x_i$ so that it it assigned across many points $y_j$ (or vice-versa). As a result, the $i$-th line (or $j$-th column) of a solution $P_\star \in \mathbb{R}_+^{n \times m}$ has usually more than one positive entry. Extending directly the formulas presented in Prop. 2 we recover extended operators that we call Kantorovich ranking and sorting operators. These operators are new to the best of our knowledge.

The K-ranking operator computes convex combinations of rank values (as described in the entries $n\overline{\mathbf{b}}$) while the K-sorting operator computes convex combinations of values contained in $\mathbf{x}$ directly. Note that we consider here convex combinations (weighted averages) of these ranks/values, according to the Euclidean geometry. Extending more generally these combinations to Fréchet means using alternative geometries (KL, hyperbolic, etc) on these ranks/values is left for future work. Because these quantities are only defined pointwisely (we output vectors and not functions) and depend on the ordering of $\mathbf{a}, \mathbf{x}, \mathbf{b}, \mathbf{y}$, we drop our reference to measure $\xi$ in notations.

**Definition 1.** *For any $(\mathbf{x}, \mathbf{a}, \mathbf{y}, \mathbf{b}) \in \mathbb{R}^n \times \Sigma_n \times \mathbb{O}_m \times \Sigma_m$, let $P_\star \in U(\mathbf{a}, \mathbf{b})$ be an optimal solution for (1) with a given convex function $h$. The K-ranks and K-sorts of $\mathbf{x}$ w.r.t to $\mathbf{a}$ evaluated using $(\mathbf{b}, \mathbf{y})$ are respectively:*

$$\widetilde{R}(\mathbf{a}, \mathbf{x}; \mathbf{b}, \mathbf{y}) := n\mathbf{a}^{-1} \circ (P_\star \overline{\mathbf{b}}) \in [0, n]^n,$$
$$\widetilde{S}(\mathbf{a}, \mathbf{x}; \mathbf{b}, \mathbf{y}) := \mathbf{b}^{-1} \circ (P_\star^T \mathbf{x}) \in \mathbb{O}_m.$$

The K-rank vector map $\widetilde{R}$ outputs a vector of size $n$ containing a continuous rank for each entry for $\mathbf{x}$ (these entries can be alternatively interpreted as $n$ times a "synthetic" CDF value in $[0, 1]$, itself a convex mixture of the CDF values $\overline{\mathbf{b}}_j$ of the $y_j$ onto which each $x_i$ is transported). $\widetilde{S}$ is a split-quantile operator outputting $m$ increasing values which are each, respectively,

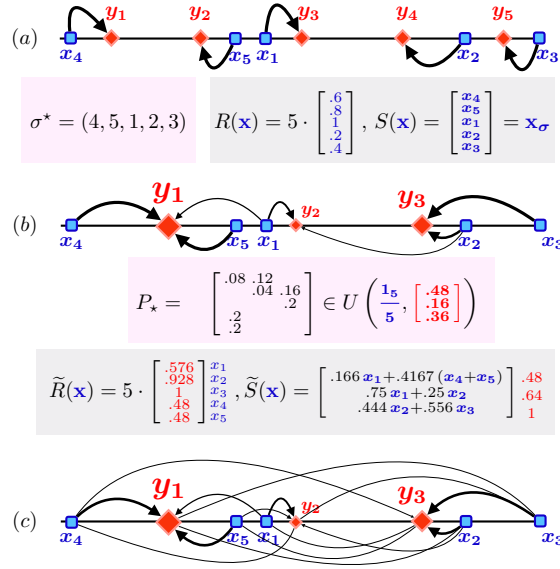

Figure 1: *(a)* sorting seen as transporting optimally $\mathbf{x}$ to milestones in $\mathbf{y}$. *(b)* Kantorovich sorting generalizes the latter by considering target measures $\mathbf{y}$ with $m = 3$ non-uniformly weighted points (here $\mathbf{b} = [.48, .16, .36]$). K-ranks and K-sorted vectors $\widetilde{R}, \widetilde{S}$ are generalizations of $R$ and $S$ that operate by mixing ranks in $\mathbf{b}$ or mixing original values in $\mathbf{x}$ to form continuous ranks for the elements in $\mathbf{x}$ and $m$ "synthetic" quantiles at levels $\overline{\mathbf{b}}$. *(c)* Entropy regularized OT generalizes further K-operations by solving OT with the Sinkhorn algorithm, which results in dense transport plans differentiable in all inputs.

barycenters of some of the entries in $\mathbf{x}$. The fact that these values are increasing can be obtained by a simple argument in which $\xi$ and $\upsilon$ are cast again as uniform measures of the same size using duplicated supports $x_i$ and $y_j$, and then use the monotonicity given by the third identity of Prop. 2.

**Computations and Non-differentiability** The generalized ranking and sorting operators presented in Def. 1 are interesting in their own right, but have very little practical appeal. For one, their computation relies on solving an OT problem at a cost of $O(nm(n + m) \log(nm))$[35] and remains therefore far more costly than regular sorting, even when $m$ is very small. Furthermore, these operators remain fundamentally *not* differentiable. This can be hinted by the simple fact that it is difficult to guarantee in general that a solution $P_\star$ to (1) is unique. Most importantly, the Jacobian $\partial P_\star / \partial \mathbf{x}$ is, very much like $R$, null almost everywhere. This can be visualized by looking at Figure 1*(b)* to notice that an infinitesimal change in $\mathbf{x}$ would not change $P_\star$ (notice however that an infinitesimal change in weights $\mathbf{a}$ would; that Jacobian would involve North-west corner type mass transfers). All of these pathologies — computational cost, non-uniqueness of optimal solution and non-differentiability — can be avoided by using regularized OT [10].

## 3 The Sinkhorn Ranking and Sorting Operators

Both K-rank $\widetilde{R}$ and K-sort $\widetilde{S}$ operators are expressed using the optimal solution $P_\star$ to the linear program in (1). However, $P_\star$ is not differentiable w.r.t inputs $\mathbf{a}, \mathbf{x}$ nor parameters $\mathbf{b}, \mathbf{y}$ [2, §5]. We propose instead to rely on a differentiable variant [10, 11] of the OT problem that uses entropic regularization [38, 17, 24], as detailed in [28, §4]. This differentiability is reflected in the fact that the optimal regularized transport plan is a dense matrix (yielding more arrows in Fig. 1*(c)*), which ensures differentiability everywhere w.r.t. both $\mathbf{a}$ and $\mathbf{x}$.

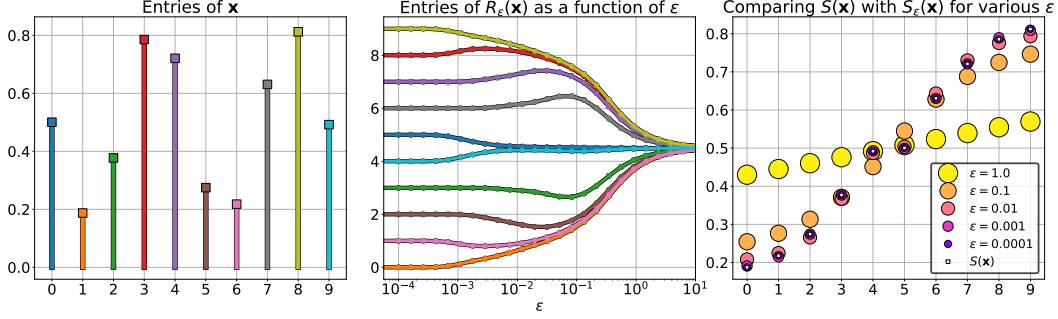

Figure 2: Behaviour of the S-ranks $\widetilde{R}_\varepsilon\left(\mathbf{a},\mathbf{x};\mathbf{b},\mathbf{y}\right)$ and S-sort operators $\widetilde{S}_\varepsilon\left(\mathbf{a},\mathbf{x};\mathbf{b},\mathbf{y}\right)$ as a function of $\varepsilon$. Here $n = m = 10$, $\mathbf{b}$ is uniform and $\mathbf{y} = (0,\ldots,m-1)/(m-1)$ is the regular grid in $[0,1]$. *(left)* input data $\mathbf{x}$ presented as a bar plot. *(center)* Vector output of $\widetilde{R}_\varepsilon\left(\mathbf{a},\mathbf{x};\mathbf{b},\mathbf{y}\right)$ (various continuous) ranks as a function of $\varepsilon$. When $\varepsilon$ is small, one recovers an integer valued vector of ranks. As $\varepsilon$ increases, regularization kicks in and produces mixtures of rank values that are continuous. These mixed ranks are closer for values that are close in absolute terms, as is the case with the 0-th and 9-th index of the input vector whose continuous ranks are almost equal when $\varepsilon \approx 10^-2$. *(right)* vector of "soft" sorted values. These converge to the average of values in $\mathbf{x}$ as $\varepsilon$ is increased.

Consider first a regularization strength $\varepsilon > 0$ to define the solution to the regularized OT problem:

$$P_\star^\varepsilon := \underset{P \in U(\mathbf{a},\mathbf{b})}{\operatorname{argmin}} \ \langle P, C_{\mathbf{xy}}\rangle - \varepsilon H(P) \quad,\text{where} \quad H(P) = -\sum_{i,j} P_{ij}\left(\log P_{ij} - 1\right) .$$

One can easily show [10] that $P_\star^\varepsilon$ has the factorized form $\mathbf{D}(\mathbf{u})K\mathbf{D}(\mathbf{v})$, where $K = \exp(-C_{\mathbf{xy}}/\varepsilon)$ and $\mathbf{u} \in \mathbf{R}^n$ and $\mathbf{v} \in \mathbf{R}^m$ are fixed points of the Sinkhorn iteration outlined in Alg. 1. To differentiate $P_\star^\varepsilon$ w.r.t. $\mathbf{a}$ or $\mathbf{x}$ one can use the implicit function theorem, but this would require solving a linear system using $K$. We consider here a more direct approach, using algorithmic differentiation of the Sinkhorn iterations, after a number $\ell$ of iterations needed for Alg. 1 to converge [19, 4, 15]. That number $\ell$ depends on the choice of $\varepsilon$ [16]: typically, the smaller $\varepsilon$, the more iterations $\ell$ are needed to ensure that each successive update in $\mathbf{v}, \mathbf{u}$ brings the column-sum of the iterate $\mathbf{D}(\mathbf{u})K\mathbf{D}(\mathbf{v})$ closer to $\mathbf{b}$, namely that the difference between $\mathbf{v} \circ K^T\mathbf{u}$ and $\mathbf{b}$ (as measured by a discrepancy function $\Delta$ as used in Alg. 1) falls below a tolerance parameter $\eta$. Assuming $P_\star^\varepsilon$ has been computed, we introduce Sinkhorn ranking and sorting operators by simply appending an $\varepsilon$ subscript to the quantities presented in Def. 1, and replacing $P_\star$ in these definitions by the regularized OT solution $P_\star^\varepsilon = \mathbf{D}(\mathbf{u})K\mathbf{D}(\mathbf{v})$.

**Definition 2** (Sinkhorn Rank & Sort). *Given a regularization strength $\varepsilon > 0$, run Alg.1 to define*

$$\widetilde{R}_\varepsilon\left(\mathbf{a},\mathbf{x};\mathbf{b},\mathbf{y}\right) := n\mathbf{a}^{-1} \circ \mathbf{u} \circ K(\mathbf{v} \circ \overline{\mathbf{b}}) \in [0,n]^n,$$

$$\widetilde{S}_\varepsilon\left(\mathbf{a},\mathbf{x};\mathbf{b},\mathbf{y}\right) := \mathbf{b}^{-1} \circ \mathbf{v} \circ K^T(\mathbf{u} \circ \mathbf{x}) \in \mathbb{R}^m.$$

**Sensitivity to $\varepsilon$.** Parameter $\varepsilon$ plays the same role as other temperature parameters in previously proposed smoothed sorting operators [29, 36, 18]: the smaller $\varepsilon$ is, the closer the Sinkhorn operator's output is to the original vectors of ranks

---

**Algorithm 1:** Sinkhorn

**Inputs:** $\mathbf{a}, \mathbf{b}, \mathbf{x}, \mathbf{y}, \varepsilon, h, \eta$
$C_{\mathbf{xy}} \leftarrow [h(y_j - x_i)]_{ij}$;
$K \leftarrow e^{-C_{\mathbf{xy}}/\varepsilon}, \mathbf{u} = \mathbf{1}_n$;
**repeat**
 $\mathbf{v} \leftarrow \mathbf{b}/K^T\mathbf{u}, \ \mathbf{u} \leftarrow \mathbf{a}/K\mathbf{v}$
**until** $\Delta(\mathbf{v} \circ K^T\mathbf{u}, \mathbf{b}) < \eta$;
**Result:** $\mathbf{u}, \mathbf{v}, K$

---

and sorted values; The bigger $\varepsilon$, the closer $P_\star^\varepsilon$ to matrix $\mathbf{a}\mathbf{b}^T$, and therefore all entries of $\widetilde{R}_\varepsilon$ collapse to the average of $n\bar{\mathbf{b}}$, while all entries of $\widetilde{S}_\varepsilon$ collapse to the weigted average (using $\mathbf{a}$) of $\mathbf{x}$, as illustrated in Fig. 2. Although choosing a small value for $\varepsilon$ might seem natural, in the sense that $\widetilde{R}_\varepsilon, \widetilde{S}_\varepsilon$ approximate more faithfully $R, S$, one should not forget that this would result in recovering the deficiencies of $R, S$ in terms of differentiability. When *learning* with such operators, it may therefore be desirable to use a value for $\varepsilon$ that is large enough to ensure $\partial P_\star^\varepsilon / \partial \mathbf{x}$ has non-null entries. We usually set $\varepsilon = 10^{-2}$ or $10^{-3}$ when $\mathbf{x}, \mathbf{y}$ lie in $[0,1]$ as in Fig. 2. We have kept $\varepsilon$ fixed throughout Alg. 1, but we do notice some speedups using scheduling as advocated by [34].

**Parallelization.** The Sinkhorn computations laid out in Algorithm 1 imply the application of kernels $K$ or $K^T$ to vectors $\mathbf{v}$ and $\mathbf{u}$ of size $m$ and $n$ respectively. These computation can be carried out in parallel to compare $S$ vectors $\mathbf{x}_1, \ldots, \mathbf{x}_S \in \mathbb{R}^n$ of real numbers, with respective probability weights $\mathbf{a}_1, \ldots, \mathbf{a}_S$, to a single vector $\mathbf{y}$ with weights $\mathbf{b}$. To do so, one can store all kernels $K_s := e^{-C_s/\varepsilon}$ in a tensor of size $S \times n \times m$, where $C_s = C_{\mathbf{x}_s\mathbf{y}}$.

**Numerical Stability.** When using small regularization strengths, we recommend to cast Sinkhorn iterations in the log-domain by considering the following stabilized iterations for each pair of vectors $\mathbf{x}_s, \mathbf{y}$, resulting in the following updates (with $\boldsymbol{\alpha}$ and $\boldsymbol{\beta}$ initialized to $\mathbf{0}_n$ and $\mathbf{0}_m$),

$$
\begin{aligned}
\boldsymbol{\alpha} &\leftarrow \varepsilon \log \mathbf{a} + \min_\varepsilon \left( C_{\mathbf{x}_s\mathbf{y}} - \boldsymbol{\alpha}\mathbf{1}_m^T - \mathbf{1}_n\boldsymbol{\beta}^T \right) + \boldsymbol{\alpha}, \\
\boldsymbol{\beta} &\leftarrow \varepsilon \log \mathbf{b} + \min_\varepsilon \left( C_{\mathbf{x}_s\mathbf{y}}^T - \mathbf{1}_m\boldsymbol{\alpha}^T - \boldsymbol{\beta}\mathbf{1}_n^T \right) + \boldsymbol{\beta},
\end{aligned}
\tag{3}
$$

where $\min_\varepsilon$ is the soft-minimum operator applied linewise to a matrix to output a vector, namely for $M \in \mathbf{R}^{n \times m}$, $\min_\varepsilon(M) \in \mathbf{R}^n$ and is such that $[\min_\varepsilon(M)]_i = -\varepsilon(\log \sum_j e^{-M_{ij}/\varepsilon})$. The rationale behind the substractions/additions of $\boldsymbol{\alpha}$ and $\boldsymbol{\beta}$ above is that once a Sinkhorn iteration is carried out, the terms inside the parenthesis above are normalized, in the sense that once divided by $\varepsilon$, their exponentials sum to one (they can be used to recover a coupling). Therefore, they must be negative, which improves the stability of summing exponentials [28, §4.4].

**Cost function.** Any nonnegative convex function $h$ can be used to define the ground cost, notably $h(u) = |u|^p$, with $p$ set to either 1 or 2. Another important result that we inherit from OT is that, assuming $\varepsilon$ is close enough to 0, the transport matrices $P_\varepsilon^\star$ we obtain should *not* vary under the application of any increasing map to each entry in $\mathbf{x}$ or $\mathbf{y}$. We take advantage of this important result to stabilize further Sinkhorn's algorithm, and at the same time resolve the thorny issue of being able

---

**Algorithm 2:** Sinkhorn Ranks/Sorts

**Inputs:** $(\mathbf{a}_s, \mathbf{x}_s)_s \in (\Sigma_n \times \mathbb{R}^n)^S, (\mathbf{b}, \mathbf{y}) \in \Sigma_m \times \mathbb{O}_m, h, \varepsilon, \eta, \widetilde{g}$.
$\forall s, \widetilde{\mathbf{x}}_s = \widetilde{g}(\mathbf{x}_s), C_s = [h(y_j - (\widetilde{\mathbf{x}}_s)_i)]_{ij}, \boldsymbol{\alpha}_s = \mathbf{0}_n, \boldsymbol{\beta}_s = \mathbf{0}_m$.
**repeat**
$\quad \forall s, \boldsymbol{\beta}_s \leftarrow \varepsilon \log \mathbf{b}_s + \min_\varepsilon \left( C_s^T - \mathbf{1}_m\boldsymbol{\alpha}_s^T - \boldsymbol{\beta}_s\mathbf{1}_n^T \right) + \boldsymbol{\beta}_s$
$\quad \forall s, \boldsymbol{\alpha}_s \leftarrow \varepsilon \log \mathbf{a}_s + \min_\varepsilon \left( C_s - \boldsymbol{\alpha}_s\mathbf{1}_m^T - \mathbf{1}_n\boldsymbol{\beta}_s^T \right) + \boldsymbol{\alpha}_s$
**until** $\max_s \Delta \left( \exp \left( C_{\mathbf{x}_s\mathbf{y}}^T - \mathbf{1}_m\boldsymbol{\alpha}_s^T - \boldsymbol{\beta}_s\mathbf{1}_n^T \right) \mathbf{1}_n, \mathbf{b} \right) < \eta$;
$\forall s, \widetilde{R}_\varepsilon(\mathbf{x}_s) \leftarrow \mathbf{a}_s^{-1} \circ \exp \left( C_{\mathbf{x}_s\mathbf{y}} - \boldsymbol{\alpha}_s\mathbf{1}_m^T - \mathbf{1}_n\boldsymbol{\beta}_s^T \right) \overline{\mathbf{b}}$,
$\forall s, \widetilde{S}_\varepsilon(\mathbf{x}_s) \leftarrow \mathbf{b}_s^{-1} \circ \exp \left( C_{\mathbf{x}_s\mathbf{y}}^T - \mathbf{1}_m\boldsymbol{\alpha}_s^T - \boldsymbol{\beta}_s\mathbf{1}_n^T \right) \mathbf{x}_s$.
**Result:** $\left( \widetilde{R}_\varepsilon(\mathbf{x}_s), \widetilde{S}_\varepsilon(\mathbf{x}_s) \right)_s$.

---

to settle for a value for $\varepsilon$ that can be used consistently, regardless of the range of values in $\mathbf{x}$. We propose to set $\mathbf{y}$ to be the regular grid on $[0, 1]$ with $m$ points, and rescale the input entries of $\mathbf{x}$ so that they cover $[0, 1]$ to define the cost matrice $C_{\mathbf{xy}}$. We rescale the entries of $\mathbf{x}$ using an increasing squashing function, such as arctan or a logistic map. We also notice in our experiments that it is important to standardize input vectors $\mathbf{x}$ before squashing them into $[0, 1]^n$, namely to apply, given a squashing function $g$, the map $\tilde{g}$ on $\mathbf{x}$ before computing the cost matrix $C_{\mathbf{xy}}$:

$$
\tilde{g} : \mathbf{x} \mapsto g \left( \frac{\mathbf{x} - (\mathbf{x}^T\mathbf{1}_n)\mathbf{1}_n}{\frac{1}{\sqrt{n}}\|\mathbf{x} - (\mathbf{x}^T\mathbf{1}_n)\mathbf{1}_n\|_2} \right).
\tag{4}
$$

The choices that we have made are summarized in Alg. 2, but we believe there are opportunities to perfect them depending on the task.

**Soft $\tau$ quantiles.** To illustrate the flexibility offered by the freedom to choose a non-uniform target measure $\mathbf{b}, \mathbf{y}$, we consider the problem of computing a smooth approximation of the $\tau$ quantile of a discrete distribution $\xi$, where $\tau \in [0, 1]$. This smooth approximation can be obtained by transporting $\xi$ towards a tilted distribution, with weights split roughly as $\tau$ on the left and $(1 - \tau)$ on the right, with the addition of a small "filler" weight in the middle. This filler weight is set to a small value $t$, and is designed to "capture" whatever values may lie close to that quantile. This choice results in $m = 3$, with

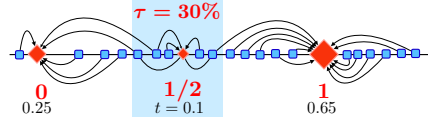

Figure 3: Computing the 30% quantile of 20 values as the weighted average of values that are selected by the Sinkhorn algorithm to send their mass onto filler weight $t$ located halfway in $[0, 1]$, and "sandwiched" by two masses approximately equal to $\tau, 1 - \tau$.

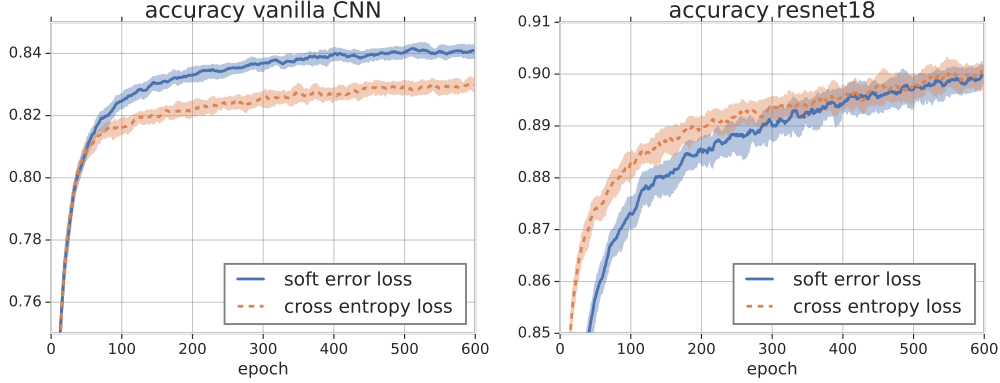

Figure 4: Error bars (averages over 12 runs) for test accuracy curves on CIFAR-10 using the same networks structures, a vanilla CNN for 4 convolution layers on the left and a resnet18 on the right. We use the ADAM optimizer with a constant stepsize set to $10^{-4}$.

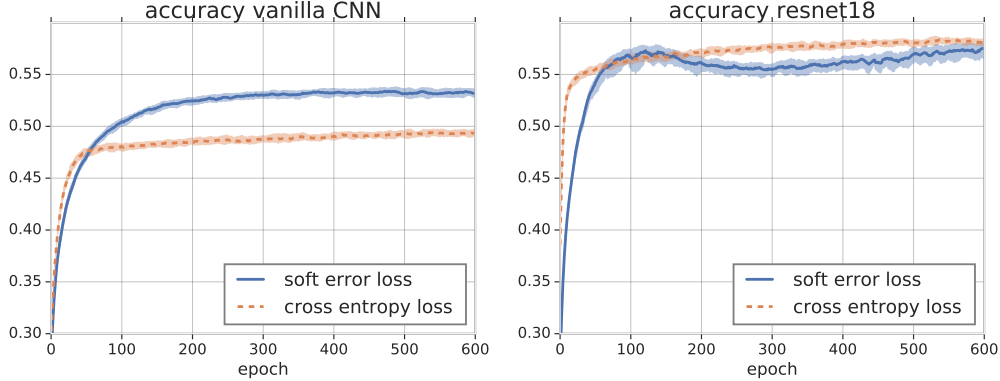

Figure 5: Identical setup to Fig. 4, with the CIFAR-100 database.

weights $\mathbf{b} = [\tau - t/2, t, 1 - \tau - t/2]$ and target values
$\mathbf{y} = [0, 1/2, 1]$ as in Figure 3, in which $t = 0.1$. With
such weights/locations, a differentiable approximation to the $\tau$-quantile of the inputs can be recovered as the second entry of vector $\tilde{S}_\varepsilon$:

$$\tilde{q}_\varepsilon(\mathbf{x}; \tau, t) = \left[ \widetilde{S}_\varepsilon \left( \frac{\mathbf{1}_n}{n}, \mathbf{x}; \begin{bmatrix} \tau - t/2 \\ t \\ 1 - \tau - t/2 \end{bmatrix}, \begin{bmatrix} 0 \\ \frac{1}{2} \\ 1 \end{bmatrix}, h \right) \right]_2. \tag{5}$$

## 4  Learning with Smoothed Ranks and Sorts

**Differentiable approximation of the top-$k$ Loss.** Given a set of labels $\{1, \dots, L\}$ and a space $\Omega$ of input points, a parameterized multiclass classifier on $\Omega$ is a function $f_\theta : \Omega \to \mathbb{R}^L$. The function decides the class attributed to $\omega$ by selecting a label with largest activation, $l^\star \in \mathrm{argmax}_l [f_\theta(\omega)]_l$. To train the classifier using a training set $\{(\omega_i, l_i)\} \in (\Omega \times \mathcal{L})^N$, one typically resorts to minimizing the cross-entropy loss, which results in solving $\min_\theta \sum_i \mathbf{1}_L^T \log f_\theta(\omega_i) - [f_\theta(\omega_i)]_{l_i}$.

We propose a differentiable variant of the 0/1 and more generally top $k$ losses that bypasses combinatorial consideration [27, 39] nor builds upon non-differentiable surrogates [6]. Ignoring the degenerate case in which $l^\star$ is not unique, given a query $\omega$, stating that the the label $l^\star$ has been selected is equivalent to stating that the entry indexed at $l^\star$ of the vector of ranks $R(f_\theta(\omega))$ is $L$. Given a labelled pair $(\omega, l)$, the 0/1 loss of the classifier for that pair is therefore,

$$\mathcal{L}_{0/1}(f_\theta(\omega), l) = H\left(L - [R(f_\theta(\omega))]_l\right), \tag{6}$$

| algorithm | n=3 | n=5 | n=7 | n=9 | n=15 |
|---|---|---|---|---|---|
| Stochastic NeuralSort | 0.920 (0.946) | 0.790 (0.907) | 0.636 (0.873) | 0.452 (0.829) | 0.122 (0.734) |
| Deterministic NeuralSort | 0.919 (0.945) | 0.777 (0.901) | 0.610 (0.862) | 0.434 (0.824) | 0.097 (0.716) |
| Our | **0.928 (0.950)** | **0.811 (0.917)** | **0.656 (0.882)** | **0.497 (0.847)** | **0.126 (0.742)** |

Table 1: Sorting exact and partial precision on the neural sort task averaged over 10 runs. Our method performs better than the method presented in [18] for all the sorting tasks, with the exact same network architecture.

where $H$ is the heaviside function: $H(u) = 1$ if $u > 0$ and $H(u) = 0$ for $u \leq 0$. More generally, if for some labelled input $\omega$, the entry $[R(f_\theta)]_{l_o}$ is bigger than $L - k + 1$, then that labelled example has a top-$k$ error of 0. Conversely, if $[R(f_\theta)]_l$ is smaller than $L - k + 1$, then the top-$k$ error is 1. The top-$k$ error can be therefore formulated as in (6), where the argument $L - [R(f_\theta(\omega)]_l$ within the Heaviside function is replaced by $L - [R(f_\theta(\omega)]_l - k + 1$.

The 0/1 and top-k losses are unstable on two different counts: $H$ is discontinuous, and so is $R$ with respect to the entries $f_\theta(\omega)$. The differentiable loss that we propose, as a replacement for cross-entropy (or more generalized top-$k$ cross entropy losses [1]), leverages therefore both the Sinkhorn rank operator and a smoothed Heaviside like function. Because Sinkhorn ranks are always within the boundaries of $[0, L]$, we propose to modify this loss by considering a continuous increasing function $J_k$ from $[0, L]$ to $\mathbb{R}$:

$$\widetilde{\mathcal{L}}_{k,\varepsilon}(f_\theta(\omega), l) = J_k \left( L - \left[ \widetilde{R}_\varepsilon \left( \frac{\mathbf{1}_L}{L}, f_\theta(\omega); \frac{\mathbf{1}_L}{L}, \frac{\overline{\mathbf{1}}_L}{L}, h \right) \right]_l \right),$$

We propose the simple family of ReLU losses $J_k(u) = \max(0, u - k + 1)$, and have focused our experiments on the case $k = 1$. We train a vanilla CNN (4 Conv2D with 2 max-pooling layers, ReLU activation, 2 fully connected layers, batchnorm on each) and a Resnet18 on CIFAR-10 and CIFAR-100. Fig. 4 and 5 report test-set classification accuracies / epochs. We used $\varepsilon = 10^{-3}$, $\eta = 10^{-3}$, a squared distance cost $h(u) = u^2$ and a stepsize of $10^{-4}$ with the ADAM optimizer.

**Learning CNNs by sorting handwritten numbers.** We use the MNIST experiment setup in [18], in which a CNN is given $n$ numbers between between 0 and 9999 given as 4 concatenated MNIST images. The labels are the ranks (within $n$ pairs) of each of these $n$ numbers. We use the code kindly made available by the authors. We use 100 epochs, and confirm experimentally that S-sort performs on par with their neural-sort function. We set $\varepsilon = 0.005$.

**Least quantile regression.** The goal of least quantile regression [32] is to minimize, given a vector of response variables $z_1, \ldots, z_N \in \mathbb{R}$ and regressor variables $\mathbf{W} = [\mathbf{w}_1, \ldots, \mathbf{w}_N] \in \mathbb{R}^{d \times N}$, the $\tau$ quantile of the loss between response and predicted values, namely writing $\mathbf{x} = (|z_i - f_\theta(\mathbf{w}_i)|)_i$ and setting $\mathbf{a} = \mathbf{1}_N / N$ and $\xi$ the measure with weights $\mathbf{a}$ and support $\mathbf{x}$, to minimize w.r.t. $\theta$ the quantile $\tau$ of $\xi$.

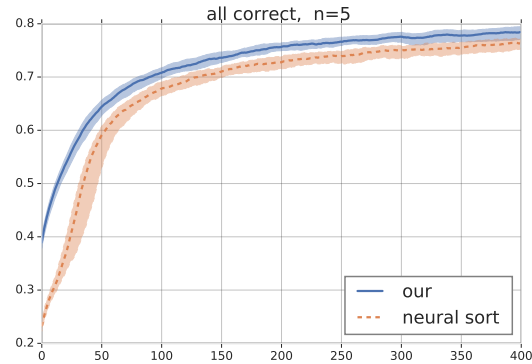

Figure 6: Test accuracy on the simultaneous MNIST CNN / sorting task proposed in [18] (average of 12 runs)

We proceed by drawing mini-batches of size 512. Our baseline method (labelled $\varepsilon = 0$) consists in identifying which point, among those 512, has an error that is equal to the desired quantile, and then take gradient steps according to that point. Our proposal is to consider the soft $\tau$ quantile $\tilde{q}_\varepsilon(\mathbf{x}; \tau, t)$ operator defined in (5), using for the filler weight $t = 1/512$. This is labelled as $\varepsilon = 10^{-2}$. We use the datasets considered in [31] and consider the same regressor architecture, namely a 2 hidden layer NN with hidden layer size 64, ADAM optimizer and steplength $10^{-4}$. Results are summarized in Table2. We consider two quantiles, $\tau = 50\%$ and $90\%$.

For each quantile/dataset pair, we report the original (not-regularized) $\tau$ quantile of the errors evaluated on the entire training set, on an entire held-out test set, and the MSE on the test set of the

| Quantile | $\tau = 50\%$ | | | | | | $\tau = 90\%$ | | | | | |
|---|---|---|---|---|---|---|---|---|---|---|---|---|
| Method | $\varepsilon = 0$ | | | $\varepsilon = 10^{-2}$ (our) | | | $\varepsilon = 0$ | | | $\varepsilon = 10^{-2}$ (our) | | |
| Dataset | Train | Test | MSE | Train | Test | MSE | Train | Test | MSE | Train | Test | MSE |
| bio | 0.33 | 0.31 | 0.83 | **0.28** | **0.28** | **0.81** | 1.17 | 1.19 | **0.74** | **1.15** | **1.18** | 1.17 |
| bike | 0.23 | **0.46** | **0.82** | **0.14** | 0.49 | 0.87 | 0.76 | 1.60 | 0.65 | **0.69** | **1.57** | **0.63** |
| facebook | **0.00** | **0.01** | **0.18** | 0.04 | 0.04 | 0.19 | **0.21** | 0.27 | 0.27 | 0.27 | 0.27 | **0.22** |
| star | 0.55 | **0.68** | **0.80** | **0.33** | 0.74 | 0.89 | 1.29 | **1.55** | 0.77 | **1.15** | 1.57 | 0.77 |
| concrete | 0.35 | **0.45** | **0.58** | **0.25** | 0.51 | 0.61 | 0.83 | 1.08 | **0.50** | **0.72** | 1.08 | 0.51 |
| community | 0.27 | **0.30** | **0.48** | **0.06** | 0.32 | 0.53 | 0.77 | 0.98 | 0.46 | **0.56** | 0.98 | **0.44** |

Table 2: Least quantile losses (averaged on 12 runs) obtained on datasets compiled by [31]. We consider two quantiles, at 50% and 90%. The baseline method ($\varepsilon = 0$) consists in estimating the quantile empirically and taking a gradient step with respect to that point. Our method ($\varepsilon = 10^{-2}$) uses the softquantile operator $\tilde{q}_\varepsilon(\mathbf{x}; \tau, t)$ defined in (5), using for the filler weight $t = 1/512$. We observe better performance at train time (which may be due to a "smoothed" optimization landscape with less local minima) but different behaviors on test sets, either using the quantile loss or the MSE. Note that we report here for both methods and for both train and test sets the "true" quantile error metric.

function that is recovered. We notice that our algorithm reaches overall better quantile errors on the training set—this is our main goal—but comparable test/MSE errors.

**Conclusion.** We have proposed in this paper differentiable proxies to the ranking and sorting operations. These proxies build upon the existing connection between sorting and the computation of OT in 1D. By generalizing sorting using OT, and then introducing a regularized form that can be solved using Sinkhorn iterations, we recover the simple benefit that all of its steps can be easily automatically differentiated. We have shown that, with a focus on numerical stability, one can use there operators in various settings, including smooth extensions of test-time metrics that rely on sort, and which can be now used as training losses. For instance, we have used the Sinkhorn sort operator to provide a smooth approximation of quantiles to solve least-quantile regression problems, and the Sinkhorn rank operator to formulate an alternative to the cross-entropy that can mimic the $0/1$ loss in multiclass classification. This smooth approximation to the rank, and the resulting gradient flow that we obtain is strongly reminiscent of *rank based dynamics*, in which players in a given game produce an effort (a gradient) that is a direct function of their rank (or standing) within the game, as introduced by [7]. Our use of the Sinkhorn algorithm can therefore be interpreted as a smooth mechanism to enact such dynamics. Several open questions remain: although the choice of a cost function $h$, target vector $\mathbf{y}$ and squashing function $g$ (used to form vector $\tilde{x}$ in Alg. 1, using Eq. 4) have in principle no influence on the vector of Sinkhorn ranks or sorted values *in the limit when $\varepsilon$ goes to* 0 (they all converge to $R$ and $S$), these choices strongly shape the differentiability of $\tilde{R}_\varepsilon$ and $\tilde{S}_\varepsilon$ when $\varepsilon > 0$. Our empirical findings suggest that whitening and squashing all entries within $[0, 1]$ is crucial to obtain stable numerically, but more generally to retain consistent gradients across iterations, without having to re-define $\varepsilon$ at each iteration.

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
