[Reviews · NeurIPS 2019]

Reviewer 1



Post-response comments: In light of great enthusiasm from other reviewers, I see no reason to stand in the way of acceptance. I was already leaning positive on this paper, so will bump my rating to a 7. I was glad that the authors provided more convincing experimental results in their response, and that they agreed to provide more examples to accompany their verbal descriptions and equations. These will help to prevent obfuscation of the simple and attractive generalization that they present in the paper. Related to this: I think they misinterpreted my point here, and suggested a change to the introduction that I don't think is necessary. I was fine with their introduction as is. I think that grasping the basic idea of their generalization is well within the capability of the average NIPS reader, if presented in a concrete manner. Lastly, I might suggest that they briefly add notational notes on their use of \circ for elementwise multiplication and vector notation for permutations. ============================================= I found the paper to be a nice theoretical discussion of generalized sorting, quantiles, and CDFs. It includes some experiments that show its applicability, but it was not clear to me that these were effective uses of the concepts constructed in the first part of the work. I'm interested to hear other opinions on this. As such, I tend very slightly towards acceptance, with a recommendation to work on finding more effective uses for the tools presented, if rejection is the result. More below: Originality: I think the basic idea of generalizing sorting is quite fundamental and interesting to consider, so I quite liked this part. As acknowledged by the authors, there seem to be a few other works that have targeted differentiability of related concepts via Sinkhorn, so I'm less certain on the degree of novelty for this portion. The applications did not seem overly original. Lastly, I look forward to hearing the perspective of others since I'm not intimately familiar with works [1,11,15,16]. Quality: The theoretical portion is quite complete, but I did not feel like the experimental results were terribly supportive of its efficacy in the suggested uses. Clarity: The paper was mostly clear, but I found the notation to be a little confounding. The use of function composition as elementwise product in Def. 1 was a particularly unfortunate use, IMO. The greatest contribution to my understanding of the concept was given by the single example of Fig. 1, so I applaud its inclusion. Further small notes in the Improvements section later. Significance: As noted, high on the theoretical front, low on the experimental front, as far as I can tell.

Reviewer 2



# Basic idea is cool I think the basic idea is to do stuff like this: def compute_differentiable_quantile(Xs, quant, epsilon, n_iterations, metric): P = sinkorn(Xs, [0,1,2 ..., len(Xs)], epsilon, n_iterations,metric) return (Xs @ P) [int(quant * (len(Xs)-1))] Note that this is differentiable w.r.t X. What's more, as long as the metric is translation-invariant, it turns out you get exact quantile computation in the limit as epsilon ->0 and n_iterations -> infinity. I suppose as epsilon -> infinity you'll get the mean of the Xs. This is cool. I don't think you needed to motivate it as much as you did, first describing the exact OT and proving the connection to quantiles and then loosening it to sinkorn. Instead, if you just write the procedure down and then say "hey look this is a quantile when epsilon -> 0" and prove in the appendix, everyone will be just as happy. I actually found the motivation distracting, I'm afraid. Anyway, idea is cool. # A question left undiscussed So your set \mathbb{O}_n seems really important. In the exact OT case I see that it doesn't matter, but in Sinkhorn land it seems like your choice could be really important. Maybe I'm missing something, but I'd like to see this discussed. # Results Not great, but that's ok. This thing seems so general that its got to have a use somewhere.

Reviewer 3



After reading the feedback from the authors: I am pleased to see that the authors have already managed to obtain much more convincing numerical results. I maintain my rating, and look forward to other applications of the Sinkhorn sorting operator in the future. == The article starts with an equivalence between sorting and an optimal transport problem. It leverages this equivalence to introduce a new generalized sorting operator, called Kantorovich sorting, and an entropy-regularized version called Sinkhorn sorting. The latter is special in that it corresponds to a differentiable function of its input vector, as opposed to standard sorting. Consequently some uses of sorting operators in machine learning can then benefit from gradients, which would be unavailable with the standard sorting operator. The paper introduces new fundamental objects, Kantorovich and Sinkhorn sorting operators (as well as CDFs and quantile functions), in a very insightful and elegant presentation (despite a number of typos, see below). Applications to a variety of tasks, such as top-k loss, quantile regression, or soft quantile normalization, help to understand the potential use for machine learning. Nevertheless, the main contribution is of a conceptual nature, and while it is hard to anticipate where exactly the proposed objects will be most useful, they are clearly of interest on their own right, and very innovative. The numerical experiments are a nice addition but it seems clear that the most fruitful applications of the new operators are yet to be found. There are also some open questions about the choice of "m" and of the regularization parameter. Overall I find the proposed manuscript to be very exciting. It is very original and very clear thanks to examples and diagrams. It can have far-reaching implications for a vast number of tasks in machine learning, despite the relatively modest advantages illustrated in the numerical experiments. There are a number of typos that can be easily fixed.

[Author Response · NeurIPS 2019]

We thank all reviewers for their encouraging comments, and we apologize for many typos, notably in Def.2. Reviewers
have all expressed similar reservations about experiments: We address these first and follow next with detailed answers.
Experiments: We made an important finding when re-considering the default settings we set for the various parameters
of our operators: $m, \mathbf{b}, \mathbf{y} \in \mathbb{O}_m$ (l.128), $\varepsilon, \ell$ (l.181), cost $h$ (l.207) and rescaling of input values (l.214). Although
all of these choices converge to standard sort (regardless of $\mathbf{y}$, $h$ or of the rescaling) as soon as $n = m, \mathbf{a} = \mathbf{b}$ and
$\varepsilon \to 0$, these choices do impact the "shape" of the differentiability encoded in our operators when $\varepsilon > 0$ (see p.6). Our
finding is that our initial choice to do a **min-max** or a **softmax rescaling** of the input arrays into $[0, 1]$ as in l.214 was
**not a good choice**, leading to **instabilities and "squeezed" values**. Applying instead a **logistic map** on **standardised**
**input values** (in batchnorm fashion) leads to **improved results across the board** (see below): Vanilla CNNs trained
on CIFAR-10/100 with the soft-error loss (l.248) beat on average those trained with XE; RESNETs yield comparative
results with both losses; In regression, using the soft-quantile loss (Eq.4) yields SOTA results when minimizing median
error ($\tau = 0.5$), but mixed for small $\tau = 0.1, 0.2$ (current understanding: soft-quantiles are computed on mini-batches,
therefore our loss is differentiable but *biased*, while the pinball loss is *not*-differentiable but *unbiased*). Finally, we beat
neuralsort [11] on the setup shared in their colab. **We are now ready to share our code, for others to build upon**.

CIFAR 10 (vanilla - resnet), see Fig. 4    CIFAR100 (vanilla - resnet), see Fig. 4    , Fig. 5

Quantile Regression Experiments (Table 1 in draft)

| quantile | method | bio | bike | facebook | star | concrete | community |
|---|---|---|---|---|---|---|---|
| $\tau = 0.1$ | S-Quantile | 0.211 | 0.171 | 0.018 | 0.354 | 0.478 | 0.201 |
| | Pinball | **0.109** | **0.144** | 0.018 | **0.166** | **0.103** | **0.095** |
| $\tau = 0.2$ | S-Quantile | 0.220 | **0.191** | 0.036 | 0.363 | 0.433 | 0.204 |
| | Pinball | **0.206** | 0.271 | 0.036 | **0.277** | **0.180** | **0.174** |
| $\tau = 0.5$ | S-Quantile | **0.241** | **0.267** | 0.090 | **0.382** | **0.297** | **0.205** |
| | Pinball | 0.430 | 0.514 | 0.090 | 0.419 | 0.302 | 0.336 |

Neural Sort MNIST Task (Fig. 5 in draft, Table 1 in [11])

| algorithm | Stoc. NS | Determ. NS | Ours |
|---|---|---|---|
| $n = 3$ | 0.920 (0.946) | 0.919 (0.945) | **0.928 (0.950)** |
| $n = 5$ | 0.790 (0.907) | 0.777 (0.901) | **0.811 (0.917)** |
| $n = 7$ | 0.636 (0.873) | 0.610 (0.862) | **0.656 (0.882)** |
| $n = 9$ | 0.452 (0.829) | 0.434 (0.824) | **0.497 (0.847)** |
| $n = 15$ | 0.122 (0.734) | 0.097 (0.716) | **0.126 (0.742)** |

Reviewer #1: ▶ *applications did not seem overly original [...] were not the most natural.* We picked these tasks because
they are representative of what can be done when switching to a rank/quantile based perspective. As envisioned by R2
and R3, we also believe many original applications will follow. ▶ *not intimately familiar with works [1,11,15,16] [...]*
*very little about [1,15,16]* We will clarify this difference. These papers use Sinkhorn to define a differentiable mechanism
to produce a bistochastic matrix (as a relaxed permutation) from high-dimensional inputs, using a parameterized matrix-
valued function (*e.g.* $\mathbf{A}$ to $\mathcal{Z}^i(\mathbf{A})$ in p.6 of [1]). We use Sinkhorn + the maths of OT in 1D to define soft-CDFs and
soft-quantiles **vector outputs**, as defined in Def.2. ▶ *the experimental results were [not] terribly supportive* We hope
our new results are more convincing. ▶ *[...] helpful to give an example* We will work out an entire example in a
blog post. ▶ *equations were typically far more complex than the simple ideas involved.* We respectfully disagree: the
flexibility of our framework requires this level of detail, notably when $\mathbf{a} \neq \mathbf{b}$ (see for example l.234). Note that these
equations are exactly what we implemented in our code. We do however agree that the paper is currently hard to digest
for casual Neurips readers.In order not to overwhelm them with technical details, we will move all of the OT related
material from the intro in §1 to §2. Instead, we will introduce directly our work as "black-box" differentiable plug-ins
for sorting, and encapsulate all the OT discussion in §2,3. ▶ *[...] top row (uniform, presumably?)* You are right.
▶ *Typo for $\tilde{Q}^l$ in Def. 2* You are right. A transpose $^T$ and a $\circ$ are also missing before and after $K$. Apologies.
Reviewer #2: ▶ *I think the basic idea is to do stuff like this [...].* Your pseudo-code agrees with the spirit of our work.
However, we argue that our implementation is far more effective, owing to the flexibility given by weights $\mathbf{b}$ and
number of targets $m$. Writing $n =$`len(Xs)` and $\ell =$`n_iterations`, your implementation requires $n^2 \cdot \ell$ operations.
Ours has linear complexity $n \cdot m \cdot \ell$ (see l.225). When computing a single quantile—the case you consider in your
pseudo-code—we bring $m$ down to 3 (see l.234, and bottom row of Fig.3 where $m = 5$). Finally, we integrate directly
`quant` within $\mathbf{b}$, no need to round to get an index (see Figs. 2 & 1(b) ). ▶ *I actually found the motivation distracting.*
As mentioned to R1, we have moved references to OT from §1 to §2,3. The rigorous derivation of our tools is now
encapsulated in §2,3, which can be skipped on a first read. We will "sell" in §1 our operators as "no-brainer" plug-in
replacements to regular sort. ▶ $\mathbb{O}_n$ *seems really important [...] like to see this discussed.* Indeed, it is crucial that
$\mathbf{y} \in \mathbb{O}_m$ to recover the soft-ranks/sorts of $\mathbf{x}$. This is discussed in l.93-97 (we will expand). This is the "magic" from OT
theory that makes everything work! (magic revealed in the proof of [Theo 2.9,23] ▶ *Results. Not great, but that's ok.*
We hope our new results are more convincing. ▶ *Some like algebra, some like code.* Agreed, we will add more code.
Reviewer #3: ▶ *More convincing numerical experiments would certainly improve the paper.* We are excited to report
more convincing experiments. ▶ *[...] considered certain algorithms that involve sorting as intermediate steps.* Your
point illustrates perfectly our motivation and we are genuinely excited by the opportunities given by this new tool. For
instance, quantile-based losses are related to fairness (e.g. arxiv 1907.08646) and we are keen to investigate connections.

[Meta-Review · NeurIPS 2019]

Sorting is fundamental to many ML tasks. This paper establishes a nice connection between sorting and optimal transport to derive a generalization of sorting called Kantorovich sorting. It adds entropy regularization to get to a version they call Sinkhorn sorting which turns out to be differentiable in the argument being sorted. Authors are requested to make the changes promised in their rebuttal before submitting the final version.